# Hepatoprotective Effect of Kombucha Tea in Rodent Model of Nonalcoholic Fatty Liver Disease/Nonalcoholic Steatohepatitis

**DOI:** 10.3390/ijms20092369

**Published:** 2019-05-13

**Authors:** Chanbin Lee, Jieun Kim, Sihyung Wang, Sumi Sung, Namgyu Kim, Hyun-Hee Lee, Young-Su Seo, Youngmi Jung

**Affiliations:** 1Department of Integrated Biological Science, Pusan National University, 63-2 Pusandaehak-ro, Geumjeong-gu, Pusan 46241, Korea; lcb102@pusan.ac.kr (C.L.); jieun@pusan.ac.kr (J.K.); s.wang@pusan.ac.kr (S.W.); sungsm06@pusan.ac.kr (S.S.); titanic620@pusan.ac.kr (N.K.); ehyuna0210@naver.com (H.-H.L.); yseo2011@pusan.ac.kr (Y.-S.S.); 2Department of Microbiological Sciences, Pusan National University, 63-2 Pusandaehak-ro, Geumjeong-gu, Pusan 46241, Korea; 3Department of Biological Sciences, Pusan National University, 63-2 Pusandaehak-ro, Geumjeong-gu, Pusan 46241, Korea

**Keywords:** Kombucha tea, nonalcoholic steatohepatitis, hedgehog, lipotoxicity

## Abstract

Kombucha tea (KT) has emerged as a substance that protects the liver from damage; however, its mechanisms of action on the fatty liver remain unclear. Therefore, we investigated the potential role of KT and its underlying mechanisms on nonalcoholic fatty liver disease (NAFLD). *db*/*db* mice that were fed methionine/choline-deficient (MCD) diets for seven weeks were treated for vehicle (M + V) or KT (M + K) and fed with MCD for four additional weeks. Histomorphological injury and increased levels of liver enzymes and lipids were evident in the M + V group, whereas these symptoms were ameliorated in the M + K group. The M + K group had more proliferating and less apoptotic hepatocytic cells than the M + V group. Lipid uptake and lipogenesis significantly decreased, and free fatty acid (FFA) oxidation increased in the M + K, when compared with the M + V group. With the reduction of hedgehog signaling, inflammation and fibrosis also declined in the M + K group. Palmitate (PA) treatment increased the accumulation of lipid droplets and decreased the viability of primary hepatocytes, whereas KT suppressed PA-induced damage in these cells by enhancing intracellular lipid disposal. These results suggest that KT protects hepatocytes from lipid toxicity by influencing the lipid metabolism, and it attenuates inflammation and fibrosis, which contributes to liver restoration in mice with NAFLD.

## 1. Introduction

Non-alcoholic fatty liver disease (NAFLD) is one of the most common liver diseases in the world. NAFLD has emerged as a public health problem, as it is closely linked with obesity and insulin resistance, diabetes mellitus, and other metabolic diseases [1]. NAFLD consists of a spectrum that ranges from simple steatosis to non-alcoholic steatohepatitis (NASH), the latter of which is characterized by massive hepatocyte death, inflammation, and fibrosis in at least one-quarter of NASH patients [2]. NASH is correlated with the systemic disorder of lipid homeostasis [3]. Steatosis occurs because of an imbalance between lipid import and removal in the liver [4]. The inflow of dietary fatty acid to the liver and de novo lipogenesis, which is the biochemical process of synthesizing fatty acid from acetyl-CoA, are major influx in the NASH [5]. The elimination of hepatic fatty acid by beta-oxidation and the exporting of very-low-density lipoprotein (VLDL) is reduced in NASH [6]. The excessive accumulation of lipids is synthesized into triglyceride (TG), which is the most evident form of lipid in the fatty liver [7]. TG itself is not toxic, per se, but the inhibition of TG synthesis increases the accumulation of fatty acids, which are more hepatotoxic [8]. These excessive fatty acids cause mitochondria dysregulation and promote the recruitment of inflammatory cytokines, such as tumor necrosis factor alpha (TNFα) [9,10,11]. Inflammation accompanies the hepatic fibrotic response, which progresses to cirrhosis and hepatocellular carcinoma [12]. Therefore, reducing the lipotoxicity that is caused by the accumulation of free fatty acid is an important strategy in the treatment of NASH.

Hedgehog (Hh) signaling is activated when the liver sustains damage, which regulates the repair response [13]. In NASH, massive hepatocyte death is observed, and these dying hepatocytes release Hh ligands that initiate the proliferation of Hh-responsive cells, such as progenitors and hepatic stellate cells (HSCs) [14]. Ballooned hepatocytes, which are a characteristic of NASH, produce Sonic hedgehog (Shh), one of Hh ligands [15,16,17]. These apoptotic hepatocytes release Shh and Indian Hh (Ihh), which trigger Hh signaling in progenitors and HSCs, eventually contributing to liver fibrosis [18]. These findings explain the relationship of the fibrosis stage with hepatocyte ballooning in NASH, which shows that Hh signaling is an essential signaling pathway involved in the pathogenesis of NASH.

Kombucha tea (KT) is a slightly sweet, acidic tea beverage, which is currently consumed around the world. It is a fermented tea that is generated by the symbiotic association of bacteria and yeasts, called “tea fungus” [19]. Recently, the protective effects of KT on several organs have been reported [19,20,21,22]. Specifically, KT has been shown to improve the levels of liver function–related enzymes and markers in livers that are damaged by acetaminophen [20]. In addition, Aloulou et al. demonstrated that KT has anti-hyperglycemic and anti-lipidemic effects, as it suppresses the activities of α-amylase and pancreatic lipase in diabetes alloxan-induced rats [21]. In a previous study, we showed that KT attenuated hepatic lipid accumulation and damage in *db*/*db* mice with acute liver injury [22]. These findings underline the protective effects of KT on the liver. However, the mechanism that underlies the protective role of KT in the liver is poorly understood. Moreover, it remains unclear whether and how KT has a curative effect upon NASH. Herein, we investigate the effects of KT in livers of mice that were fed with methionine/choline-deficient (MCD) for 11 weeks, and find that KT ameliorated hepatic damage in these mice by increasing hepatocyte survival and decreasing inflammation and fibrosis through the reduction of Hh activation. In addition, KT exerted protective action in hepatocytes from lipotoxin by suppressing lipid accumulation. These findings suggest that KT has the potential to prevent or treat NAFLD and NASH.

## 2. Results

### 2.1. KT Attenuates Hepatic Damage Caused by MCD Diet in db/db Mice

To examine the effects of KT in NASH, we generated NASH-like animal models by feeding *db*/*db* mice with an MCD diet for seven weeks; then, for an additional four weeks, the mice were treated with either KT (M + K) or vehicle (M + V), while continuing the MCD diet (Figure 1). Hematoxylin and eosin (H&E) staining showed fatty hepatocytes in *db*/*db* mice (control mice: CON). The MCD diet led to the great accumulation of fatty hepatocytes with macro- and microvesicular lipid droplets in *db*/*db* mice, and KT reduced macrovesicular lipid droplet-contained hepatocytes (Figure 2a). Both liver and body weight decreased in MCD-fed *db*/*db* mice when compared with chow-fed *db*/*db* mice. Interestingly, KT treatment increased body weight in MCD-fed *db*/*db* mice, although the ratio of liver weight to body weight (LW/BW) was not significant different between the two MCD-fed groups (Appendix A). The serum level of alanine transaminase (ALT) was significantly elevated in the two MCD-fed groups when compared with the CON group, but it was higher in the M + V group than in the M + K group (CON: 14.24 ± 2.94; M + V: 66.50 ± 4.28; M + K: 51.15 ± 2.86; * *p* < 0.05; ** *p* < 0.005). The aspartate transaminase (AST) level in the M + V group significantly increased when compared with other two groups (CON: 43.09 ± 9.63; M + V: 76.96 ± 5.43; M + K: 49.44 ± 3.74; * *p* < 0.05; ** *p* < 0.005) (Figure 2b). The amount of hepatic TG was higher in the MCD-fed group than the chow-fed group, but KT significantly reduced the level of hepatic TG in the livers of mice that were treated with MCD as compared with livers of MCD-fed mice. In addition, MCD treatment downregulated the RNA level of the glucose-6-phosphatase catalytic subunit (*G6pc*), which is one of the glycolysis enzymes in the liver, in both groups of MCD-fed *db*/*db* mice compared with the CON *db*/*db* mice. However, KT upregulated *G6pc* expression in the M + K group as compared with the M + V group. These data revealed that KT reduced the hepatic fatty level and attenuated liver damage in MCD-fed *db*/*db* mice.

### 2.2. KT Improves Hepatocyte Survival

Given that apoptotic hepatocytes greatly increase in NASH [23] and that KT improved the liver function and histomorphology in MCD-fed *db*/*db* mice (Figure 2), we examined whether KT protected hepatocytes from apoptosis, contributing to liver restoration in these mice. Immunostaining for active Caspase-3, which is a marker of apoptosis [23,24], and Ubiquitin, a marker of ballooned cells [25,26], showed a greater accumulation of Caspase-3- and Ubiquitin-positive hepatocytic cells in the M + V group than in the CON and M + K groups (Figure 3). The number of cell expressing Ki-67, which is a marker of the S phase [27], significantly decreased in the M + V group when compared with other two groups. However, these hepatocytic damage was changed in the M + K group, displaying more proliferating and less apoptotic hepatocytic cells compared with the M + V group. These data suggest that KT protected hepatocytes from injury and promoted hepatocyte survival, contributing to the successful restoration of lipid-damaged livers.

### 2.3. KT Reduces Lipotoxicity by Decreasing Hepatic FFA Level in MCD-Treated db/db Mice

MCD diets increased the level of peroxisome proliferator-activated receptor gamma (*Pparγ*), which is a transcriptional factor regulating lipid metabolism [28], and specifically free fatty acid (FFA) formation by uptaking dietary FFA into the liver [5], although cluster of differentiation 36 (*Cd36*), which is another marker of FFA uptake into the liver [29], was not significantly changed in comparison with chow-fed *db*/*db* mice. However, KT significantly reduced the expression of both *Cd36* and *Pparγ* in MCD-fed mice. Diacylglycerol-*O*-acyltransferase 2 (*Dgat2*), an enzyme that is involved in TG synthesis [30], was significantly downregulated in MCD-fed mice when compared with chow-fed mice, but KT tended to upregulate its expression in the M + K as compared with the M + V group. De novo lipogenesis, which is one of the sources of FA, is the metabolic pathway that converts excess carbohydrates into FAs, eventually forming TG [31]. In several studies, FA synthesis is reported to be increased in NAFLD, but not in NASH [32,33]. In line with these findings, markers of *de novo* FA synthesis, fatty acid synthase (*Fas*)*,* and sterol regulatory element binding protein 1c (*Srebp1c*), were reduced in the M + V group when compared with the CON group. Their expression was also lower in the M + K than the M + V group. The levels of markers that were related to beta-oxidation discarding FFA, peroxisome proliferator-activated receptor gamma coactivator 1 alpha (*Ppargc1α*), fatty acid binding protein (*Fabp1*), acyl coenzyme A oxidase 1 (*Acox1),* carnitine palmitoyltransferase 1 *(Cpt1*), peroxisome proliferator-activated receptor alpha *(Pparα*), and medium-chain-acyl-coenzyme A dehydrogenase (*Mcad)* were significantly lower in the M + V than the CON group. However, mice in the M + K group significantly increased the expression of these genes that are involved in beta-oxidation as compared mice in the M + V group (Figure 4a). These data indicate that *db*/*db* mice fed with an MCD diet for 11 weeks have the dysregulated lipid metabolism, leading to increased hepatic FFA, as occurs in NASH, and treatment with KT alleviated the abnormal lipid metabolism by decreasing the uptake (*Cd36* and *Pparγ*) and synthesis of FFA (*Pparγ*, *Fas*, and *Srebp1c*) and by increasing the beta-oxidation. In addition, the tendency of a higher level of *Dgat2* in the M + K group than the M + V group indicated that KT effectively reduced lipotoxicity by reducing the accumulation of FFA in the liver. In line with these findings, the hepatic level of FFA in the M + K group tended to be lower than that of the M + V group, and it was similar to the FFA level in the CON group (Figure 4b). These findings suggest that KT is involved in lipid metabolism and discard lipotocixity in the liver.

### 2.4. KT Reduces Inflammation and Fibrosis in the Liver of db/db Mice

We investigated whether hepatic inflammation emerged in the liver of MCD-fed *db*/*db* mice and whether KT attenuated the inflammation in these mice because inflammation has been implicated in the pathogenesis of NASH [12,34]. The RNA levels of inflammation markers, such as tumor necrosis factor alpha (TNFα), interleukin-1β *(Il-1β*), chemokine (C-X-C motif) ligand 1 (*Cxcl1*), and *Cxcl2* significantly increased in the M + V group, whereas their expression decreased in the M + K group and it was nearly basal level observed in the CON group (Figure 5a). These results revealed that KT had an anti-inflammatory effect on the NASH-like mouse model.

Given that inflammation accompanies fibrosis and that fibrosis is also observed in NASH [4,12], fibrosis in the MCD-fed *db*/*db* mice treated with KT was examined. The expression of the fibrogenic markers transforming growth factor beta (*Tgfβ*), alpha-smooth muscle actin (*α-SMA*), collagen type1 α 1 (*Col1α1*), connective tissue growth factor (*Ctgf*), and tissue inhibitor of metalloproteinase 1 (*Timp1*) was higher in the M + V group than the CON and M + K group, as assessed by qRT-PCR (Figure 5b). Although the RNA level of *α-SMA* tended to be elevated in the M + V group when compared with other two groups, the protein amount of α-SMA remarkably increased in the M + V group as compared with the CON and M + K group (Figure 5c). In line with the α-SMA increase in livers of *db*/*db* mice that were fed with MCD diet, Sirius Red staining showed more collagen deposition in these mice when compared with other two groups (Figure 5d). Biochemical analysis of the hepatic hydroxyproline content, a quantitative measure of liver fibrosis, confirmed that MCD-fed *db*/*db* mice had more liver fibrosis than the CON *db*/*db* and M + K *db*/*db* mice (Figure 5e). These data suggest that KT alleviated hepatic fibrosis in MCD-fed *db*/*db* mice.

### 2.5. KT Suppresses Activation of Hh Signaling Pathways in Diet-Induced NASH

Hh signaling pathway is known to be involved in NASH progression [14,18]. In the current study, we showed that elevated hepatic inflammation and fibrosis in MCD-fed *db*/*db* mice was downregulated in these mice after KT treatment. Hence, we examined whether KT influenced the Hh expression in *db*/*db* mice that were fed with a MCD diet. Although the RNA expressions of *Shh,* an Hh ligand, and *Gli2*, an Hh-target gene, were not significantly different between the chow-fed *db*/*db* and MCD-fed *db*/*db* mice, KT treatment remarkably reduced their expressions in these mice. The expressions of *Smo*, an Hh receptor, were significantly higher in the M + V group than in the CON group, and its RNA level was lower in the M + K group than in any of the other groups (Figure 6a). Parallel to the mRNA data, immunostaining of Shh and Gli2 confirmed a greater accumulation of Shh- or Gli2-positive cells in the livers of MCD-fed *db*/*db* mice when compared with those of the other two groups (Figure 6b). Therefore, these data indicated that KT suppressed Hh expression, attenuating liver fibrosis in MCD-fed *db*/*db* mice.

### 2.6. KT Prevents Palmitate-Induced Steatosis in Primary Hepatocytes

We modelled lipotoxicity in vitro by culturing hepatocytes in the presence of palmitic acid (PA), which is a known FA toxicity inducing agent [35,36], with or without KT, to assess whether KT improved hepatocyte viability by reducing lipotoxicity. An accumulation of lipid droplets was evident in PA-treated AML 12 cells (PA group) at 12 and 24 h, whereas these lipid droplets in the PA-treated cells with KT (PA + KT group) apparently declined at 12 h and they were rarely detected 24 h post–KT treatment (Appendix Aa), as examined by Oil-red O staining. In addition, cell viability significantly decreased in PA-treated AML12 cells at 12 and 24 h, while KT significantly increased the viability of PA-treated cells. Interestingly, KT enhanced cell viability when compared with any other groups under non-PA treated conditions at 12 h (Appendix Ab). We examined the effect of KT on primary hepatocytes isolated from mice after confirming that KT lowers lipid accumulation in and improves the viability of AML12 cells. PA increased lipid accumulation, followed by decreased cell viability in primary hepatocytes (Figure 7a). KT alleviated lipid accumulation and increased cell viability in PA-treated hepatocytes (Figure 7b). To check whether the increased cell viability and decreased lipid accumulation in the PA+KT group resulted from reduced lipotoxicity, the expression of the genes that are associated with lipid metabolism was assessed in these cells. One of the FA uptake markers assessed by qRT-PCR, *Cd36*, was significantly lower after 12 h in the hepatocytes of the PA+KT group than it was in those of the PA group. Although *Fas* expression, a marker of de novo FA synthesis, significantly increased in the KT group when compared with the PA+V group, the level of *Dgat2*, a TG synthesis marker, was higher in the PA+KT group than any other groups at 12 h. In addition, elevated levels of *Ppargc1α*, *Acox1*, and *Cpt1*, which are markers of beta-oxidation, in the PA+KT group confirmed that KT attenuated lipotoxicity by lowering FFA level in PA-treated hepatocytes (Figure 8). As lipotoxic damage that is caused by FFA accumulation in hepatocytes is known to trigger Shh expression [15,16], we assessed the level of Shh in these cells. PA greatly enhanced the RNA level of *Shh* when compared with any other groups, whereas KT significantly reduced *Shh* expression in PA-given cells. Therefore, these findings suggest that KT increases TG synthesis and beta-oxidation and reduces lipotoxic injury and Shh production, which contributes to the enhanced viability of hepatocytes.

## 3. Discussion

The present study reveals the effect of KT in an experimental NASH model. In the previous research, *db*/*db* mice that were fed with MCD for one week were additionally treated with MCD for three weeks in parallel with KT; in these mice, KT reduced both lipid accumulation and liver damage [22]. Based on these findings, we fed the *db*/*db* mice with MCD for a total of eleven weeks, additionally giving them KT for the last four of these weeks, to investigate the effect of KT on a NASH-like experimental rodent model. In this model, increased lipid accumulation, especially of TG and FFA; fibrosis; and, inflammation with hepatic damage were observed in the MCD-fed *db*/*db* mice, which suggests that a NASH-like animal model was successfully generated by our system. Using this model, we demonstrated that KT effectively decreases the intracellular TG content and ALT/AST serum levels. Hepatocyte death, fibrosis, and inflammation were alleviated in KT-given *db*/*db* mice that were fed with MCD. In addition, KT suppressed lipid accumulation in damaged hepatocytes by improving lipid metabolism and, accordingly, promoted viability by removing lipotoxicity. These hepatocyte-protective functions of KT demonstrated the in vivo function of KT in the NASH-like animal model. Therefore, these findings suggest that KT has the potential effect of stabilizing chronic diseases, instead turning them into mild diseases.

Feeding *db*/*db* mice MCD diets is known to cause them to develop inflammation and fibrosis in addition to simple steatosis, and is therefore a well-recognized small-animal model for progressive obesity-related NAFLD [37,38]. An MCD diet impairs mitochondrial beta-oxidation and leads to the induction of alcohol-inducible CYP2E1 expression [37,39]. Lipotoxic stress also induces hepatocyte apoptosis and reduced cell proliferation, which destroys liver function [40]. Hence, hepatocyte death rate is an important factor accelerating the development of hepatic steatosis into NASH [23,24]. KT enhanced hepatocyte proliferation and suppressed the apoptosis of these cells (Figure 3 and Figure 6). In an analysis of gene expression that is related to lipid metabolism, we showed that KT treatment decreased FFA uptake to the liver and *de novo* lipogenesis, and increased both TG synthesis and FFA oxidation, even though MCD caused damage to the mitochondrial beta-oxidation. Cpt1, which is the mitochondrial gateway for FA entry into the matrix, is the main controller of hepatic mitochondrial beta-oxidation [41]. However, the association of Cpt1 with the NASH progression is still controversial. Perez-Carreras et al. reported that reduced the activity of Cpt1 was rarely detected in NASH patients [42]. In contrast, Nakamuta et al. demonstrated that the expression of Cpt1 was remarkably decreased in NAFLD [43]. However, other studies reported on the reduction of *Cpt1* in *db*/*db* mice that were fed with MCD [44,45]. Even though MCD injured mitochondrial beta-oxidation, KT improved FFA oxidation, contributing to the discarding of FFA in the livers of mice with NASH.

The TG level has been the basis for grading the severity of steatosis in NAFLD, although it is known to be not toxic to the liver [7,8]. In current study, MCD diet caused excessive accumulation of TG in both MCD-treated groups when compared with the CON group; KT lowered its level in the MCD-fed mice when compared with the MCD-fed mice that did not receive KT treatment. However, Dgat2, which has an essential role in synthesizing TG [30], was downregulated in both MCD-fed groups. The expression of *Dgat2* also tended to be higher in the M + K group than in the M + V group. This could be explained by lipid availability. The levels of genes that were involved in beta-oxidation were greatly downregulated in the M + V group as compared with other two groups, and the FFAs available to convert TG were plentiful. However, a lower level of *Dgat2* in the M + V group indicated that, even though the FFAs were abundant, they did not effectively convert into TG. In the KT-treated mice fed with MCD, FFA uptake was lower, and even uptaken hepatic FFAs were actively removed by beta-oxidation. The FFAs that were available for converting TG were poor when compared with those of the M + V group. Hence, even though the level of *Dgat2* was higher in the M + K group than in the M + V groups, the available amount of FFA for TG synthesis was lower in the M + K group than in the M + V groups. In line with this explanation, the chow-fed *db*/*db* mice contained higher levels of *Dgat2* with a lower amount of TGs when compared with the MCD-treated groups. A lower hepatic amount of FFA in the M + K group than in the M + V group supports this possibility (Figure 4b). In line with our findings, it has been reported that TG synthesis declines and FFA accumulation is enhanced in NASH [30]. In addition, steatosis severity, as graded by TG level, is not known to predict hepatic damage, including inflammation and fibrosis [46,47]. The accumulation of lipids other than TG causes lipotoxicity in hepatocytes [7]. Our data demonstrated that KT removes the first hit, which leaves the liver vulnerable to the second hit, which suppresses disease progression.

We exposed hepatocytes to PA to induce lipid accumulation and then gave them KT to confirm that KT protects hepatocytes from lipotoxic stress by influencing lipid metabolism. After determining the optimal concentration of PA using AML12, primary hepatocytes that were isolated from wide-type mice were treated with PA. As expected, PA-treated hepatocytes contained a great accumulation of FFA with decreased viability (Figure 7 and Appendix A). One of the markers involved in the uptake of FFA into the liver, *Cd36*, tended to be upregulated in PA-treated cells. However, hepatocyte death that was caused by PA was significantly suppressed, and the lipids stored in hepatocytes were gradually eliminated, in KT-treated cells. Increased *Cd36* in PA-exposed cells was significantly downregulated, to nearly baseline levels, in KT-treated cells with PA exposure. Expressions of genes related to beta-oxidation were rarely changed in the PA-given cells, whereas their levels were significantly enhanced in KT-given cells with PA treatment when compared with other groups, which suggested that KT effectively removed the lipids accumulated in hepatocytes. Although *Fas* and *Srebp1c* were upregulated in the PA+KT group, these genes are known to convert excess carbohydrates into FFA to make TG [48]. A higher level of *Dgat2* in the PA + KT than in the PA + V group supports the hypothesis that FFAs formed through de novo lipogenesis are effectively converted into TG. In addition, both KT-treated groups, the PA + KT and KT group, showed the increase of *Fas* mRNA. Due to glucose being one of predominant components in KT [49,50,51], it is possible that hepatocytes uptake glucose provided by KT, and increase of glucose-derived acetyl-CoA, a substrate for de novo FA synthesis [52,53,54] contributes to the upregulation of *Fas* in KT-treated hepatocyte with or without PA. Therefore, these data support that KT has a therapeutic effect on the hepatocytes of NASH by enhancing intracellular lipid disposal.

Inflammation and fibrosis are pathological characteristics that present the progression from simple steatosis to steatohepatitis, and they are significantly involved in the generation of hepatic damage [2,18]. Hepatocyte death brings about both inflammation and fibrosis [55]. The massive death and severe injury of hepatocytes is more evident in NASH than it is in simple steatosis [23], which helps to explain why patients with NASH easily develop cirrhosis [47]. Hepatocytes that are injured by lipid toxicity undergo apoptosis, and dying hepatocytes release several cytokines, such as PDGF, CTGF, TGFβ, and Hh ligands, which activate immune cells, progenitors, and hepatic stellate cells (HSCs), which are Hh-responsive [56,57,58,59]. Ballooned hepatocytes in patients with NASH produce Shh, which trigger the activation of neighboring Hh-responsive cells [15,16,25]. The activated cells turn on their Hh signaling and amplify the Hh signaling in a manner of autocrine, which eventually proliferate and/or activate in response to the Hh signaling [60]. Specifically, the HSCs stimulated by Hh signaling undergo activation and change their phenotype into myofibroblast-like cells, which are a major source of extracellular matrix protein production and contribute to further inflammation and fibrosis [61,62]. Therefore, hepatocyte protection from lipotoxic injury is an important strategy in preventing and treating NASH [18]. Growing evidence suggests that ballooned hepatocytes, which are a distinct property of NASH, produce Shh [15,16,25]. In line with these findings, we showed that Shh-positive cells were apparent in the M + V group, but these cells were rarely detected in the M + K group (Figure 6). In addition, the higher level of *Shh* in damaged hepatocytes containing accumulated lipid droplets significantly declined in KT-given hepatocytes that are injured by PA (Figure 8). These findings support the hypothesis that protecting hepatocytes from lipotoxicity using KT reduces Hh activation, as this mitigated fibrosis and inflammation in KT-treated *db*/*db* mice that were fed with MCD. In addition, Pparα has been reported to attenuate liver steatosis and inflammation in NASH [63], and agonists for Pparα have been shown to have therapeutic potential for NASH [64]. In line with these findings, *Pparα* was greatly downregulated in the livers of MCD-fed mice, while KT significantly enhanced its expression in these mice. Hence, it is also possible that KT plays an important role in restoring the livers of these mice through interaction with Pparα. Further studies are required to investigate the association of KT with these signaling pathways that are involved in NASH.

In conclusion, we demonstrated that KT suppressed lipid accumulation in hepatocytes by decreasing FFA uptake and increasing both beta-oxidation and TG synthesis, and protected hepatocytes from lipotoxicity. This protective function of KT was also in vivo verified through improved lipid metabolism and reduced fibrosis and inflammation in the livers of MCD-fed *db*/*db* mice, which led to the alleviation of liver damage. Therefore, these findings suggest that KT has potential as a therapeutic dietary supplement to prevent NASH progression or to stabilize chronic fatty liver diseases into mild fatty liver diseases.

## 4. Materials and Methods

### 4.1. Preparation of KT Extract

Six grams of black tea (type of tea bag, Lipton, Yellow Label Tea) was added to 600 mL boiling water and was infused for 5 min. and then cooled to room temperature. 10% (*w*/*v*) sucrose was added to it. Additionally, cooled black tea was poured into 1L glass beaker that had been previously sterilized at 121 °C for 20 min. It was then inoculated with freshly grown KT mat (purchased from Misokombu, Pusan, Korea) that had been cultured in the same medium for 14 days and 10% (*v*/*v*) of previously fermented liquid tea broth to prepare Kombucha tea. The beaker was covered with clean cheese cloth and fixed with rubber bands. The fermentation was carried out under room temperature (25 ± 3 °C) for 14 days. While kombucha tea was fermented, a new kombucha mat developed over the tea surface. The fermented tea was centrifuged at 7000× *g* for 20 min. and the supernatant was filtered by using 0.45 μm syringe filter (Minisart syringe filter, Satorious AG, Gӧttigen, Germany). Finally, the filtrate was lyophilized to dryness. It was kept at −80 °C and used further for the experiments against MCD-induced nonalcoholic steatohepatitis and palmitic acid induced lipotoxicity of the murine hepatocyte cell line.

### 4.2. Animal Experiments

Six-week-old male C57BKS *db*/*db* mice were purchased from Korea Research Institute Bioscience and Biotechnology (Daejeon, Korea). They were fed with normal diet, watered, and housed with a 12 h light-dark cycle for two weeks for adjustment. To examine the effects of KT in vivo, 8-week-old *db*/*db* mice were fed MCD diet for seven weeks, and then randomly divided into two groups, which were treated with MCD for an additional four weeks along with oral administration of water (M + V, *n* = 5) or 2g/kg of KT (M + K, *n* = 6) every day. As a control, the chow-fed mice were given with equal volumes of the vehicle (CON, *n* = 4) (Figure 1). Serum and liver tissue was collected for histological and biochemical analysis. Animal care and surgical procedures were approved by the Pusan National University Institutional Animal Care and Use Committee and carried out in accordance with the provisions of the National Institutes of Health Guide for the Care and Use of Laboratory Animals. The animal protocol used in this study was approved by the Pusan National University Institutional Animal Care and Use Committee (PNU-IACUC) for ethical procedures and scientific care (Approval Number PNU-2016-1307, dated 28 February 2017).

### 4.3. Isolation of Primary Hepatocytes and Cell Experiments

The hepatocytes were isolated from healthy livers of C57BL/6 mice (Hyochang, Dae-gu, Korea), as described previously [65,66]. Briefly, mice were anaesthetized with Zoletil50 (5 mg·kg^−1^ body weight, Virbac S.A, France) to immobilize them in the recumbent position on a treatment table, and the inferior vena cava was cannulated under aseptic conditions. Livers were perfused in situ with EGTA and collagenase (Sigma Aldrich, St. Louis, MO, USA) to disperse the cells. The primary hepatocytes were separated from nonparenchymal cells using Percoll density gradient centrifugation. As determined by Trypan Blue exclusion, cell viability was >92% in all of the experiments. Primary hepatocytes were cultured on collagen-coated six-well plates at a density of 1 × 10^5^ cells/well or 96-well plates at a density of 3 × 10^3^ cells/well in Williams’ Medium E without phenol red (Sigma-Aldrich, St. Louis, MO, USA), supplemented with 5% fetal bovine serum (FBS; Gibco, Thermo Fisher Scientific, Waltham, MA, USA), 1 μM dexamethasone, and a cocktail solution of penicillin/streptomycin (P/S), ITS + (insulin, transferrin, selenium complex, BSA, and linoleic acid), GlutaMAX™, and HEPES (Gibco, Thermo Fisher Scientific, Waltham, MA, USA). After cell attachment (approximately 4 h after plating), the culture medium was replaced with serum-free Williams’ Medium E containing 0.1 μM dexamethasone and a cocktail solution of P/S, ITS +, GlutaMAX™, and HEPES for hepatocyte maintenance.

The normal mouse hepatocyte cell line AML12 (CRL-2254; ATCC, Manassas, VA, USA) was cultured at a density of 1 × 10^5^ cells/well in Dulbecco’s modified Eagle’s medium/F-12 (DMEM/F-12, Gibco) and 1× antibiotics, along with insulin, transferrin, selenium, and dexamethasone, as per ATCC instructions, at 37 °C in a humidified atmosphere containing 5% CO_2_.

To assess the effects of KT against hepatocyte damage, the primary hepatocytes or AML12 cells were serum-starved in culture medium containing no FBS for 24 h before treatment. After starvation, the for 24 h. Before PA treatment, the AML12 cells were tested in three different concentrations (250 μM, 1 mM, 2 mM) of PA for 24 h and then 250 μM of PA was chosen to the optimal concentration for this experiment (data not shown). After the cells were washed with PBS twice, these cells were treated with or without KT (90 μg/mL) in culture medium for 12 or 24 h. These experiments were repeated at least three times.

### 4.4. Liver Histology and Immunohistochemistry

Liver specimens were fixed 10% neutral buffered formalin, embedded in paraffin, and cut into 4 μm sections. The specimens were deparaffinized, hydrated, and stained usual method with standard hematoxylin and eosin staining (H&E) to examine morphology and Sirius Red staining to assess fibrosis. For immunohistochemistry (IHC), the sections were incubated for 10 min. in 3% hydrogen peroxide to block endogenous peroxidase. Antigen retrieval was performed by heating in 10mM sodium citrate buffer (pH 6.0) for 10 min. The sections were blocked in Dako protein blocking solution (X9090; Dako Envision, Dako Corp., Carpinteria, CA, USA) for 30 min. and incubated with primary antibodies, anti-Ubiquitin (sc-8017; Santa Cruz Biotechnology, Inc., CA, USA), anti-active Caspase-3 (AF835; R&D systems, Minneaoplis, MN, USA), anti-Ki-67 (NCL-KI67-MM1; Leica Microsystems, Newcastle, UK), anti-Shh (sc-9024; Santa Cruz Biotechnology, Inc., CA, USA), anti-Gli2 (GWB-CE7858; Genway Biotech, Inc., San Diego, CA, USA), or non-immune sera to demonstrate staining specificity at 4 °C overnight. Polymer horseradish peroxidase (HRP), anti-rabbit (K4003; Dako, CA, USA), or anti-mouse (K4001; Dako, CA, USA) was used as secondary antibodies. 3,3′-Diaminobenzidine (DAB) (K3466; Dako, CA, USA) was employed for the detection procedure.

### 4.5. RNA Isolation and Quantitative RT-PCR

Total RNA was extracted with TRIZOL (Ambion^®^ by Life Technologies, Thermo Fisher Scientific, MA, USA). After assuring RNA quality and concentration, total RNA (5 μg) was used to synthesize cDNA using the SuperScript II First-strand Synthesis System (Invitrogen, Carlsbad, CA, USA), following the manufacturer’s instruction. Gene expression was evaluated by QRT-PCR analysis. The levels of mRNAs were quantified by real-time RT-PCR while using Power SYBR Green Master Mix (Applied Biosystem, Forster City, CA, USA) per the manufacturer’s specifications (Mastercycler Real-Time PCR, Eppendorf, Hamburg, Germany). Appendix A lists the sequences of primers for mice. Samples were analyzed in duplicate according to the Delta-Delta threshold (ΔΔC*t*) method. The expression values were normalized to the levels of 40S ribosomal protein 9S mRNA. All of the PCR products were directly sequenced for genetic confirmation in Macrogen Inc. (Seoul, Korea).

### 4.6. Western Blot

Total protein was extracted from cultured cells or frozen clamped liver tissue that had been stored at −80 °C. The whole tissues were homogenized in Triton lysis buffer (TLB) supplemented with protease inhibitors (Complete Mini 11 836 153 001; Roche, Indianapolis, IN, USA). Protein concentration was measured with a Pierce BCA Protein Assay Kit (Thermo Scientific, Waltham, MA, USA). Equal amounts of total protein were fractionated by polyacrylamide gel electrophoresis and then transferred to polyvinylidene difluoride (PVDF) membranes (Millipore, Darmstadt, Germany). Primary antibodies against α-SMA (A5228; Sigma Aldrich, St. Louis, MO, USA) and GAPDH (MCA4739; AbDSerotec, Oxford, UK) were used in this experiment. The membranes were developed by chemiluminescence (ATTO Corporation, Tokyo, Japan). The blots obtained from three independent experiments were scanned and a region of interest (ROI) around the band of interest was defined. The band intensities were calculated by using CS analyzer 4 program (ATTO Corporation.)

### 4.7. Cell Proliferation Assay

Cell proliferation was measured using a Cell Titer Proliferation Assay (MTS; Promega, Madison, WI, USA) according to the manufacturer’s instructions. In brief, primary hepatocytes or AML12 at a density of 3 × 10^3^ cells/well were plated in 96-well plates and treated with PA and KT as described in the above cell experiments. After treatment, 10 μL of MTS reagent was added to each well, and the plates were incubated at 37 °C in a CO_2_ incubator until the color developed. Absorbance was then measured at a wavelength of 490 nm using a Glomax multi-detection system (Promega).

### 4.8. Measurement of Triglyceride Level

Lipids were extracted from frozen clamped liver tissue that had been stored at −80℃. Total liver triglyceride levels were measured by using the Triglyceride Colorimetric Assay Kit from Cayman Chemical (Nashville, TN, USA), following the manufacturer’s specifications. Total liver free fatty acids levels were assessed spectrophotometrically using the half-micro test (11383175001; Roche, Mannheim, Germany), following the manufacturer’s specifications.

### 4.9. Hydroxyproline Assay

The hepatic hydroxyproline content of the livers was calculated by the method previously described [59,67]. Briefly, 50 mg of freeze-dried liver tissue was hydrolyzed in 6 N HCL at 110 Promega for 16 h. The hydrolysate was evaporated under vacuum and the sediment was re-dissolved in 1 mL distilled water. Samples were filtered using 0.22 μm filter centrifuge tube at 14,000 rpm for 5 min. The lysates were then incubated with 0.5 mL of chloramines-T solution, containing 1.41 g of chloramine-T dissolved in 80 mL of acetate-citrate buffer and 20 mL of 50% isopropanol for 20 min. at room temperature. To this 0.5 mL of Ehrlich’s solution, including 7.5 g of dimethylaminobenzaldehyde dissolved in 13 mL of 60% perchloric acid (Sigma Aldrich, St. Louis, MO, USA) and 30 mL of isopropanol (Sigma Aldrich, St. Louis, MO, USA), were added to the mixture, which was incubated at 65 °C for 15 min. After cooling to the room temperature, the absorbance was read at 561 nm. Hydroxyproline concentration was calculated from a standard curve that was prepared with high purity hydroxyproline (Sigma Aldrich, St. Louis, MO, USA) and divided by the weight of liver specimen that was employed in this analysis (μg hydroxyproline/g liver).

### 4.10. Measurement of AST/ALT

Serum aspartate aminotransferase (AST) and alanine aminotransferase (ALT) were measured by using ChemiLab GOT/GPT (IVD Lab Co., Korea), according to the manufacturer’s instructions.

### 4.11. Cell Quantification

To quantify the number of Ki-67-, Ubiquitin-, and Caspase-3- positive cells, 10 randomly chosen fields at 40× magnification were evaluated for each mouse. The positive cells of each protein were quantified by counting the total number of positive cells per filed and dividing by the total number of hepatocytes per field for each mouse.

### 4.12. Oil Red O Staining

The media was removed, the wells were gently rinsed with PBS, and the cells were fixed for 10 min. with 1 mL per well of 10% formalin. After fixing, the wells were washed with glass-distilled H_2_O three times for 30 s per rinse. Next, 1 mL of 100% propylene glycol (PEG) (Sigma Aldrich, St. Louis, MO, USA) was added to each well for 2 min. and then removed. Before using Oil Red O (ORO)/PEG (Sigma Aldrich, St. Louis, MO, USA), the ORO/PEG solution was prewarmed to 60 °C. ORO/PEG solution was added 1 mL per well, allowed to remain for overnight, removed, and the wells were then washed in 60% PEG for 1 min. The wells were washed in glass-distilled H_2_O and 1 mL per well hematoxylin was added for 10 min. The wells were then washed with glass-distilled H_2_O three times for 30s each, followed by a glass-distilled H_2_O rinse for 30 s.

### 4.13. Statistical Analysis

All of the results are expressed as mean ± standard error of mean (SEM). The statistical significances between the control and treated groups or subgroups were analyzed by the unpaired two-sample Student’s *t*-test or one-way analysis of variance (one-way ANOVA), followed by a post-hoc Tukey’s test. Data for in vivo effect of KT in the liver were analyzed with unpaired two-sample Student’s t-test and considered to be significant when *p*-values are <0.05. Data for in vitro effect of KT on hepatocytes were analyzed with one-way ANOVA and considered as significant when the *p*-values are <0.05. Statistical analyses were performed using SPSS statistics 20 software (Release version 20.0.0.0, IBM Corp., Armonk, NY, USA).

## Figures and Tables

**Figure 1 ijms-20-02369-f001:**
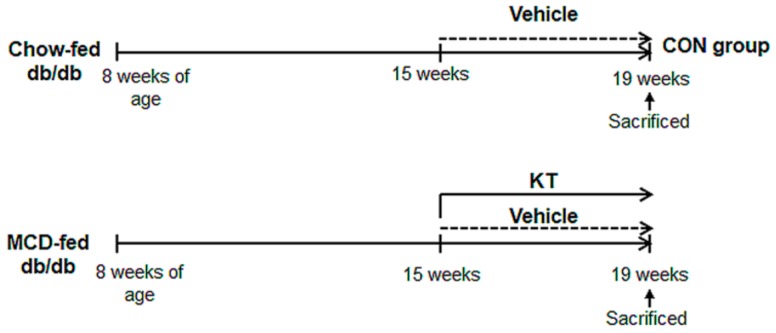
Design of mouse experimental model. After being fed with methionine/choline-deficient (MCD) for seven weeks, *db*/*db* mice were additionally treated with MCD for four weeks in parallel with oral administration of saline (M + V, *n* = 5) or Kombucha tea (KT) (M + K, *n* = 6). Chow-fed mice were treated with equal volume of vehicle (CON, *n* = 4).

**Figure 2 ijms-20-02369-f002:**
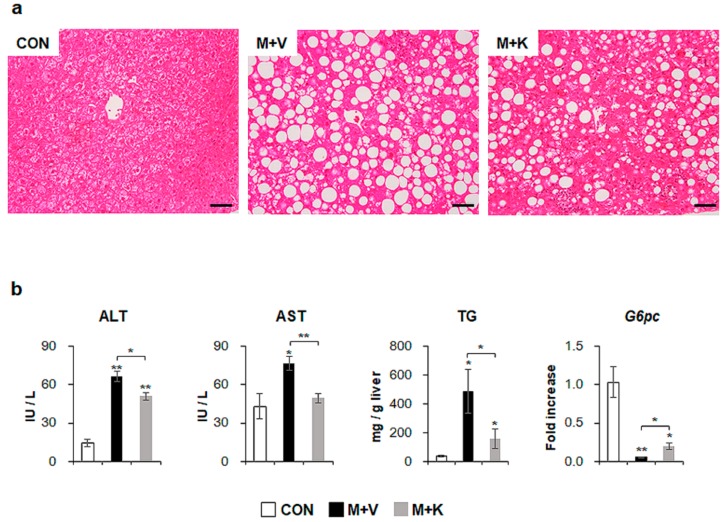
KT improves liver histomorphology and function of MCD-treated *db*/*db* mice (**a**) Hematoxylin-eosin (H&E) staining shows the representative liver morphology of each group (Scale bars: 50 μm). (**b**) Levels of alanine transaminase/aspartate transaminase (ALT/AST) in serum, hepatic triglyceride, and *G6pc* mRNA in liver of each group were graphed as mean ± SEM (*n* ≥ 3/group, * *p* < 0.05, ** *p* < 0.005 vs. control group). RNA expression of *G6pc* was expressed as fold increase.

**Figure 3 ijms-20-02369-f003:**
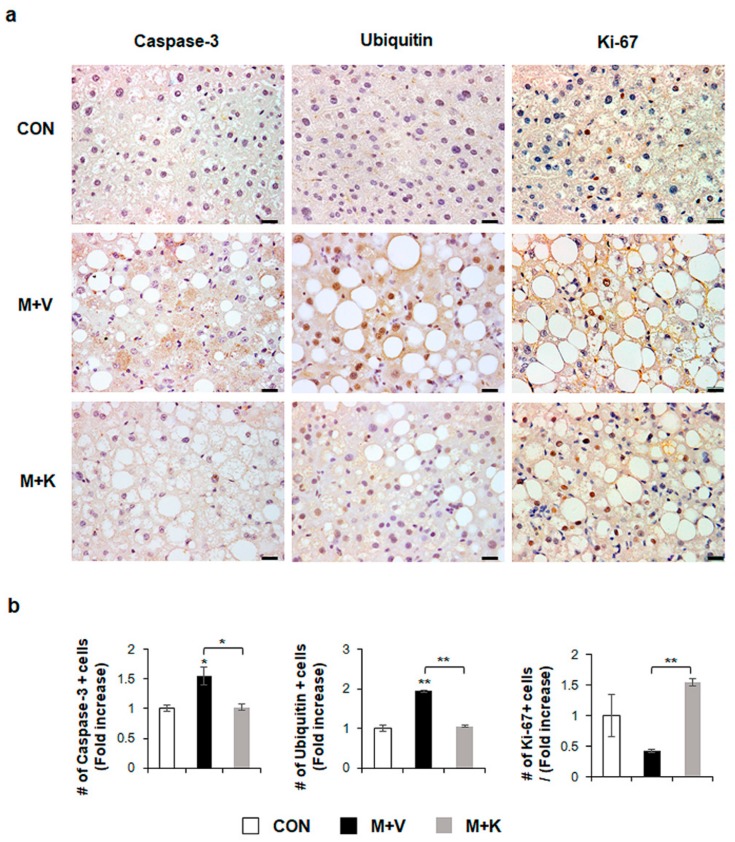
KT suppresses apoptosis and promotes proliferation of hepatocytes in MCD-fed *db*/*db* mice (**a**) Immunohistochemistry for Caspase-3, Ubiquitin, and Ki-67 for liver sections from representative mice from each group (Scale bars: 20 μm). (**b**) Quantitative Caspase-3, Ubiquitin, and Ki-67 immunohistochemistry data from all mice. The numbers of Caspase-3, Ubiquitin, or Ki-67- positive cells were counted per field and divided by the number of hepatocytes per field. Results were described as fold increase. Means ± SEM of results are graphed (*n* ≥ 3 /group, * *p* < 0.05, ** *p* < 0.005 vs. control group).

**Figure 4 ijms-20-02369-f004:**
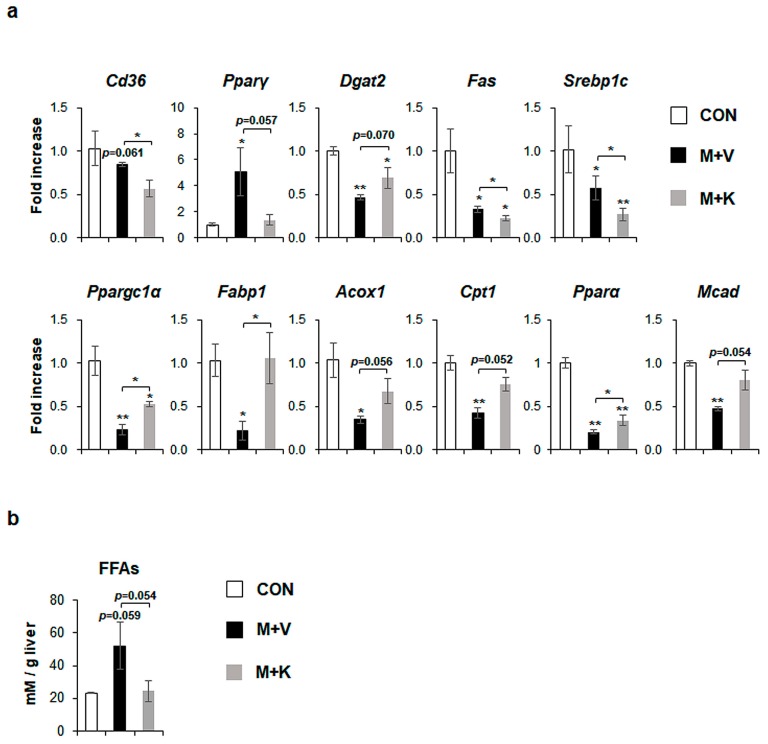
KT treatment improves hepatic lipid metabolism in MCD-fed *db*/*db* mice (**a**) qRT-PCR analysis for cluster of differentiation 36 (Cd36), peroxisome proliferator-activated receptor gamma (Pparγ), Diacylglycerol-*O*-acyltransferase 2 (Dgat2), fatty acid synthase (Fas), sterol regulatory element binding protein 1c (Srebp1c), peroxisome proliferator-activated receptor gamma coactivator 1 alpha (Ppargc1α), fatty acid binding protein (Fabp1), acyl coenzyme A oxidase 1 (Acox1), carnitine palmitoyltransferase 1 (Cpt1), peroxisome proliferator-activated receptor alpha (Pparα), and medium-chain-acyl-coenzyme A dehydrogenase (Mcad) in livers of the experimental animals. Results were described as fold increase. (**b**) Hepatic level of free fatty acids in these mice. All of the results are display as mean ± SEM (*n* ≥ 3/group, * *p* < 0.05, ** *p* < 0.005 vs. control group).

**Figure 5 ijms-20-02369-f005:**
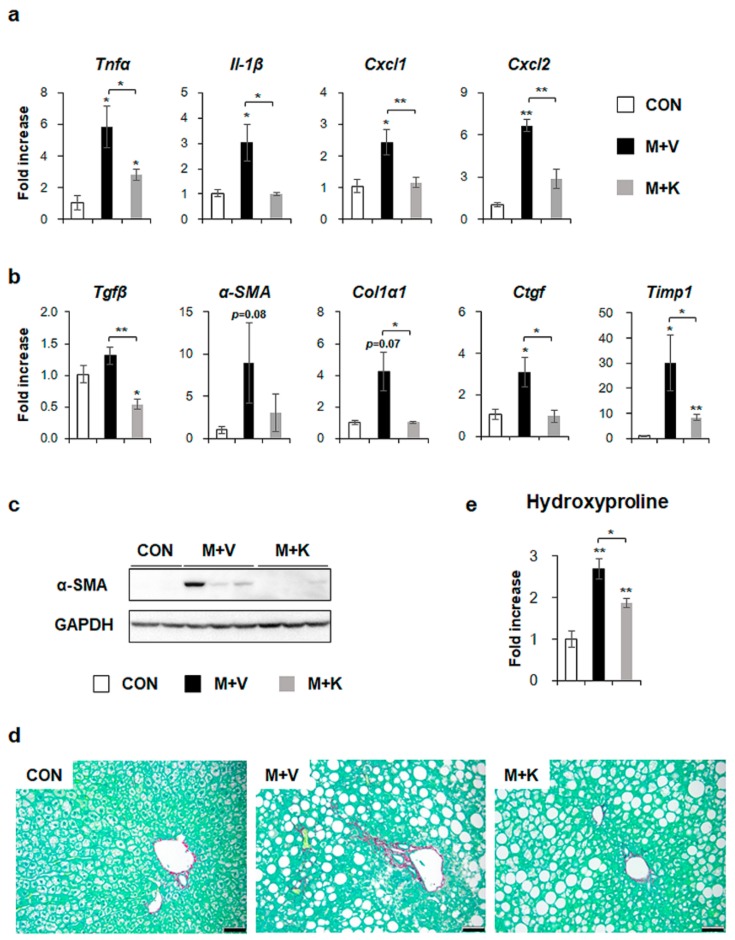
KT decreases both inflammation and fibrosis in livers of MCD-treated *db*/*db* mice (**a**) qRT-PCR analysis for tumor necrosis factor alpha (TNFα), interleukin-1β (Il-1β), chemokine (C-X-C motif) ligand 1 (Cxcl1), and Cxcl2 in livers from the CON, M + V, and M + K groups. (**b**) qRT-PCR analysis for transforming growth factor beta (Tgfβ), alpha-smooth muscle actin (α-SMA), collagen type1 α 1 (Col1α1), connective tissue growth factor (Ctgf), and tissue inhibitor of metalloproteinase 1 (Timp1). (**c**) Western blot analysis of α-SMA in liver of three representative mice from each group. GAPDH was used as internal control. Data shown represent one of three experiments with similar results. (**d**) Sirius red staining in the liver sections from representative mice from each group (Scale bar: 50 μm). (**e**) Hepatic hydroxyproline content in all mice. All of the results are display as mean ± SEM (*n* ≥ 3/group, * *p* < 0.05, ** *p* < 0.005 vs. control group).

**Figure 6 ijms-20-02369-f006:**
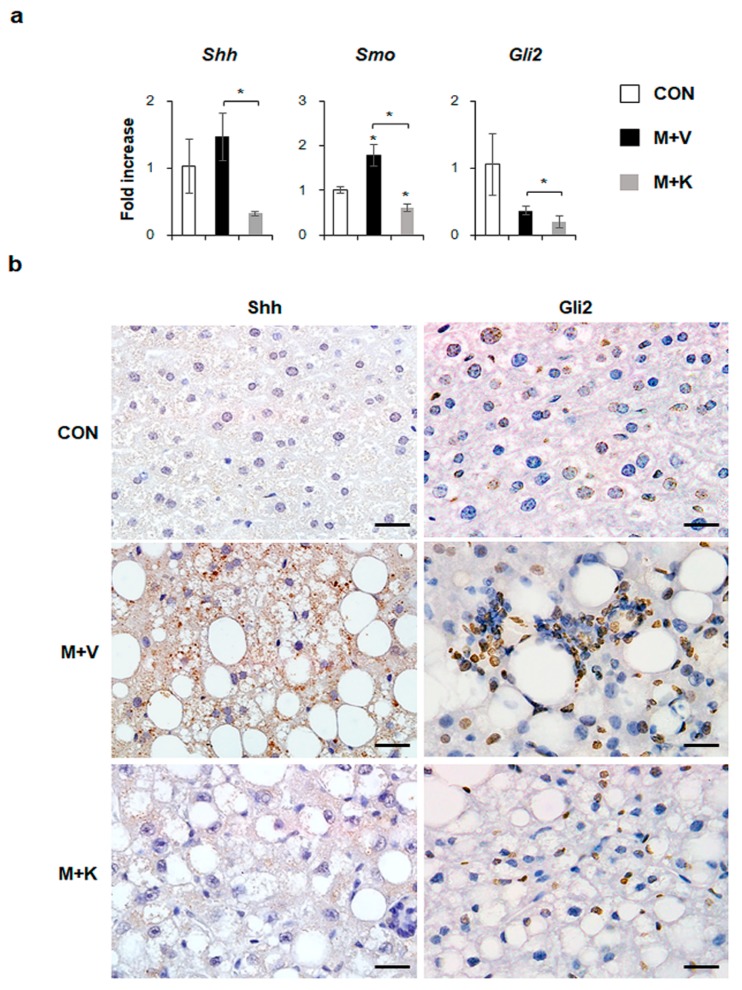
KT reduces expression of Hh pathway in *db*/*db* mice fed with MCD (**a**) qRT-PCR analysis for Sonic hedgehog (Shh), Smoothened (Smo), and GLI-Kruppel family zinc finger 2 (Gli2). Means ± SEM of results are graphed (*n* ≥ 3 /group, * *p* < 0.05, ** *p* < 0.005 vs. control group). (**b**) Immunohistochemistry for Shh and Gli2 (brown color) in the liver sections from representative mice from each group (Scale bars: 20 μm).

**Figure 7 ijms-20-02369-f007:**
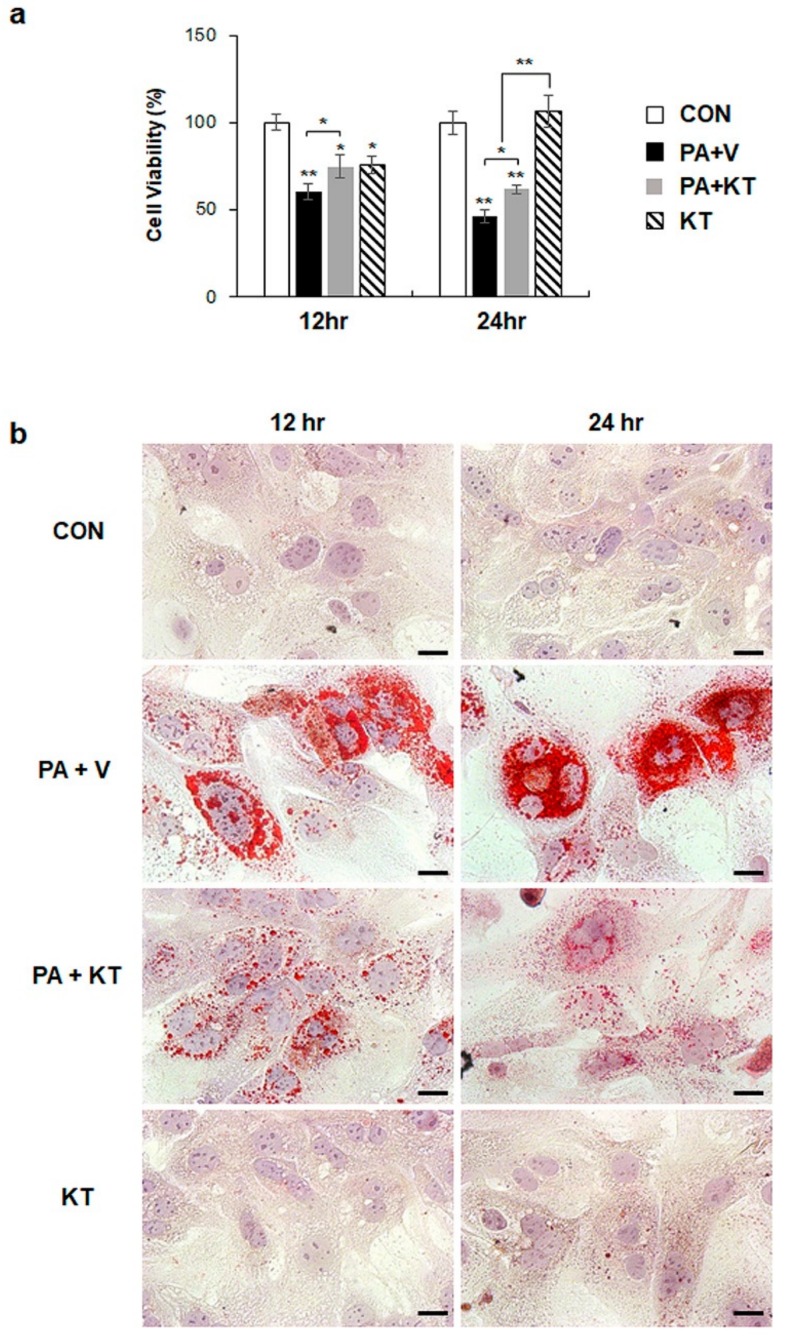
KT protects hepatocytes from palmitate-induced lipotoxicity by attenuating lipid accumulation (**a**) Cell viability of primary hepatocytes from mice treated with KT was analyzed using MTS assay. After being exposed to 250 μM of palmitic acid (PA) for 24 h, primary hepatocytes were treated with vehicle (PA + V) or KT (PA + KT) for 12 and 24 h. As a control, primary hepatocytes were treated with equal volume of vehicle (BSA) without PA for 24 h, and then given with (KT) or without KT (CON). The mean±SEM results that were obtained from three repetitive experiments are graphed (* *p* < 0.05, ** *p* < 0.005 vs. CON). (**b**) Oil red O staining for lipid droplets in these cells. Representative images are shown (Scale bars: 20 μm).

**Figure 8 ijms-20-02369-f008:**
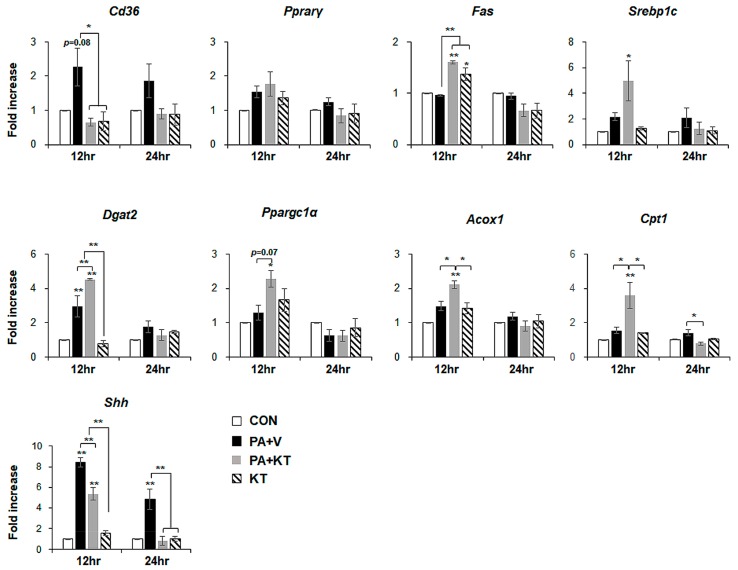
KT modulates lipid metabolism in hepatocytes damaged by PA. qRT-PCR analysis for Cd36, Pparγ, Fas, Srebp1c, Dgat2, Ppargc1α, Acox1, Cpt1, and Shh in CON, PA + V, PA + KT, and KT cell groups. The mean ± SEM results obtained from three repetitive experiments are graphed (* *p* < 0.05, ** *p* < 0.005 vs. CON group).

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
