# Peer review of "Hepatoprotective Effect of Kombucha Tea in Rodent Model of Nonalcoholic Fatty Liver Disease/Nonalcoholic Steatohepatitis"

_ijms, 2019, doi:10.3390/ijms20092369_

Round 1

Reviewer 1 Report

The present study was performed in order to investigate the potential role of KT and its underlying mechanisms on nonalcoholic fatty liver disease (NAFLD). It was found that KT increased cell proliferation and decreased apoptosis in MCD-administered db/db mice. Authors suggested that KT protects hepatocytes from lipid toxicity by increasing free fatty acid (FFA) oxidation and decreasing the lipid uptake and lipogenesis, thus, influencing the lipid metabolism, and attenuates inflammation and fibrosis, contributing to liver restoration in mice with NAFLD. The data are of interest, but there are still several questions to be addressed.

Comments

1.      Increase of cell proliferation (Ki67) by KT is a very questionable result. Promotion of cell proliferation is usually a response occurred due to the toxic effect of the substance, for example, the regenerative proliferation. Furthermore, increase of cell proliferation itself is a sign of promotion of carcinogenesis. It is very strange that MCD treatment resulted in a decrease of cell proliferation. It is necessary to provide the references on decrease of cell proliferation due to the MCD treatment from other studies. MCD itself induce injury in the liver, thus the cell proliferation it is likely to be induced.

2.      The number of mice used in vivo is very low for such type of the experiment.

3.      Authors have been using the MCD diet model of NASH. There are other NASH models, and it is important to discuss, how the present model is different from other models.

4.      Please remove repetition of Material and Methods from Results section.

5.      It is necessary to mention what kind of house-keeping gene was used in the Real-time PCR analysis (∆∆CT method).

6.      The presentation of results needs to be improved. There are too many speculations.

7.      The quality of the figures should be improved. In the downloaded file they are out of focus and with strange background under the letters.

8.      In the Page 6, line15, it is written that the protein amount of alpha-SMA remarkably increased in the M+K group. However, In the Figure 5 it is decreased. Is this a mistake?

9.      In vitro experiment it is necessary to explain and discuss why Fas expression was increased in case of KT treatment.

10.  The conclusion about the decrease of FFA amount by KT due to the increase of TG synthesis and at the same time activation of beta-oxidation is too speculative.

11.  Explanation of the Statistical analysis is poor.

12.  It is advisable to show the data, as Mean±SD, but not the Mean±SEM.

13.  The paper is written in relatively good English but still it is better to be improved by an English-speaking scientist).

Author Response

Reviewer #1

The present study was performed in order to investigate the potential role of KT and its underlying mechanisms on nonalcoholic fatty liver disease (NAFLD). It was found that KT increased cell proliferation and decreased apoptosis in MCD-administered db/db mice. Authors suggested that KT protects hepatocytes from lipid toxicity by increasing free fatty acid (FFA) oxidation and decreasing the lipid uptake and lipogenesis, thus, influencing the lipid metabolism, and attenuates inflammation and fibrosis, contributing to liver restoration in mice with NAFLD. The data are of interest, but there are still several questions to be addressed.

Comments

1. Increase of cell proliferation (Ki67) by KT is a very questionable result. Promotion of cell proliferation is usually a response occurred due to the toxic effect of the substance, for example, the regenerative proliferation. Furthermore, increase of cell proliferation itself is a sign of promotion of carcinogenesis. It is very strange that MCD treatment resulted in a decrease of cell proliferation. It is necessary to provide the references on decrease of cell proliferation due to the MCD treatment from other studies. MCD itself induce injury in the liver, thus the cell proliferation it is likely to be induced.

As you pointed out, MCD damages on the liver and liver damage means hepatocyte injury. Injured hepatocytes undergo apoptosis, and other types of cells, such as immune cells, hepatic stellate cells, and progenitors, proliferate in response to liver damage. Loss of hepatocytes means loss of liver functions and abnormal proliferation of non-parenchymal cells distort liver structure. Like other injury models, such as CCl4, TAA, and CDAHF, MCD causes hepatocyte’s death and results in proliferation of non-hepatocyte proliferation. These findings are well-known and many papers presenting the effect of MCD on hepatocytes are easily found in pubmed [Gut. 2010; 59(5): 655–665./Sci Rep. 2018; 8: 7499./ Exp Mol Med. 2017; 49(9): e380/ PLoS One. 2015; 10(5): e0127991./ Hepatology. 2017 ;66(6):1934-1951/ Am J Pathol. 2016;186(7):1762-1774, and so on]. As we described, we counted hepatocytic cells which were damaged by MCD.

PH (partial hepatectomy) model is different from these models. PH leaves 1/3 healthy hepatocytes, which proliferate in response to PH damage.

2. The number of mice used in vivo is very low for such type of the experiment.

We employed total 15 male TG mice per group (control group: 4 mice/ M+V group: 5 mice/ M+K group; 6 mice), and the number of mice used in vivo was not small. On average, we employed a similar number of mice compared to the number of experimental mice used in other groups [Hepatology. 2019 Apr 25/ Hepatology. 2014;60(1):133-45./ Gastroenterology. 2010;139(1):323-34./ Gastroenterology. 2009;137(4):1478-1488./ J Biol Chem. 2010; 285(24): 18528–18536]

3. Authors have been using the MCD diet model of NASH. There are other NASH models, and it is important to discuss, how the present model is different from other models.

As you pointed out, there are several NAFL models and it is important to discuss about them. Many papers have reviewed NAFL including animal models. In the present study, we generated the data by employing MCD-fed db/db mice model, and described the features of the MCD model focused on our findings (in discussion, from line 7 in page 10 to line 31 in page 11). Because we did not check the effect of MCD on other NASH model, discussing for the characteristics of NASH animal models in the manuscript seems to be somewhat less relevant to this research.

4. Please remove repetition of Material and Methods from Results section.

As you pointed out, we repeatedly provide the brief explanation for the experimental model, although we explained in detail about the design of animal experiments in material and methods. In results 2.1, the brief description is necessary to describe each group name. Hence, it seems to be better leave this part. However, other repeated part in result section was deleted in the revised manuscript.

5. It is necessary to mention what kind of house-keeping gene was used in the Real-time PCR analysis (∆∆CT method).

As you requested, we provided the information for house-keeping gene in the revised manuscript; “The expression values were normalized to the levels of 40S ribosomal protein 9S mRNA”.

6. The presentation of results needs to be improved. There are too many speculations.

In most papers, the last sentence in each result section summarize the finding of the results.  Hence, we also summarized the finding using verb, such as, “suggest or indicate”. Explanation for the data, such as increase or decrease of RNA & protein expression or positive cells for signals, cannot be speculated. However, some parts in result 2.7 is speculated, as you pointed out. Therefore, we deleted these parts (please see answer for comment #10). If there is more needed to be fixed, please let us know.

7. The quality of the figures should be improved. In the downloaded file they are out of focus and with strange background under the letters.

Although figures (>300dpi) in the original version of manuscript met the resolution required by IJMS journal, we uploaded figures with higher resolution (>400dpi) in the revised manuscript, as you requested.

In regard of strange background under the letters; when we checked PDF file and printed out the original version of manuscript, we did not find the strange background which you pointed out. In addition, other reviewer did not rise this concern. It seems to be technical problem. If technical problem is right, there is nothing we can do. It had better contact with the IJMS to solve this issue.

8. In the Page 6, line15, it is written that the protein amount of alpha-SMA remarkably increased in the M+K group. However, In the Figure 5 it is decreased. Is this a mistake?

It is our big mistake. As the data are presented in figure 5C, amount of alpha-SMA is higher in the M+V group than any other two groups. We corrected them in the revised manuscript; “~ the protein amount of α-SMA remarkably increased in the M+V group compared with the CON and M+K group~”.

9. In vitro experiment it is necessary to explain and discuss why Fas expression was increased in case of KT treatment.

As you requested, we provided the explanation for Fas increase in KT-treated hepatocytes in the discussion section of the revised manuscript; “In addition, both KT-treated groups, the PA+KT and KT group, showed the increase of Fas mRNA. Because glucose is one of predominant components in KT [Compr Rev Food Sci Food Saf. 2014;13(4)/ J Food Prot. 2000 Jul;63(7):976-81./ Intl. J. Food. Ferment. Technol. 6(1): 13-24, June, 2016], it is possible that hepatocytes uptake glucose provided by KT, and the increase of glucose-derived acetyl-CoA, a substrate for de novo FA synthesis [Methods Enzymol. 2015;561:197-217./Semin Liver Dis. 2015;35(3):250-61./Gastroenterology. 2008; 134(5): 1369–1375.], contributes to upregulation of Fas in KT-treated hepatocyte with or without PA”.

10. The conclusion about the decrease of FFA amount by KT due to the increase of TG synthesis and at the same time activation of beta-oxidation is too speculative.

We assessed the levels of genes-related with lipid metabolism in livers of the experimental mice, and clearly showed that KT decreased both FFA uptake to the liver and de novo lipogenesis, and increased both TG synthesis and FFA oxidation in MCD-treated db/db mice. Improved liver morphology, decreased apoptosis and increased proliferation of hepatocytes, reduced inflammation and fibrosis, alleviated Hh activation, and in vitro experimental data support that KT improves lipid metabolism and contributes to the successful liver repair or prevents disease progression. In vitro experiments, we found that Fas significantly increased and Srebp1 tended to be upregulated in the PA+KT group compared with PA+V. Based on upregulation of FFA & TG synthesis markers and reduced lipid accumulation (assessed by Oil red O staining) in KT treated cells, we suggested the possibility that KT was involved in discarding FFA by enhancing TG synthesis. As reviewer pointed out, the speculated description is not right in result section. Therefore, we deleted the overstated two speculated parts in results 2.7 of the revised manuscript.

11. Explanation of the Statistical analysis is poor.

As you requested, we added more explanation of the statistical analysis in the revised manuscript; “All results are expressed as mean ± standard error of mean (SEM). Statistical significances between control and treated groups or subgroups were analyzed by the unpaired two-sample Student’s t-test or one-way analysis of variance (one-way ANOVA) followed by a post-hoc Tukey’s test. Data for in vivo effect of KT in the liver were analyzed with unpaired two-sample Student’s t-test and considered as significant when P-values are < 0.05. Data for in vitro effect of KT on hepatocytes were analyzed with one-way ANOVA and considered as significant when P-values are < 0.05. Statistical analyses were performed using SPSS statistics 20 software (Release version 20.0.0.0, IBM Corp., Armonk, NY, USA).”

12.  It is advisable to show the data, as Mean±SD, but not the Mean±SEM.

According to articles reviewing statistical analysis in experimental biology [J Cell Biol. 2007 Apr;177(1):7-11 and BMJ. 2005 Oct 15; 331(7521): 903.], SEM is more appropriate in comparison among different groups. SD is the dispersion of data in a normal distribution. Namely, SD is used for descriptive error and indicates how accurately the mean represents sample data (average difference between the data and the mean). However SEM includes statistical inference based on the sampling, and measures how variable the mean will be. Statistician and authors in these articles explain that it is usually appropriate to show inferential error bars, such as SEM, rather than SD, to compare experimental results with controls in clinical and experimental studies. Therefore, SEM is more proper statistical analysis for our experimental data. 

13. The paper is written in relatively good English but still it is better to be improved by an English-speaking scientist).

Although you have commented on English writing in the original version of manuscript, this manuscript was proofread by professional editing company. We attached the certificate for proofreading for the first version of manuscript.

Reviewer 2 Report

The present study has demonstrated the protective effect of Kombucha tea (KT) against the both of lipid accumulation and lipotoxicity, in vivo and vitro.  Although It has been already shown the protective effect of KT against non-alcoholic fatty liver disease (NAFLD), in there, KT was shown to redress the dysbiosis induced by the methionine/choline-deficient diets.  This study has exhibited KT can intervene the hedgehog involving in non-alcoholic steatohepatitis (NASH). Why doesn't this author make the description about the hedgehog in Abstract and Discussion. The description in there will give new and distinctive impact for this manuscript.  

Therefore, this paper would be suitable for publication as an original article after making addition of the above matters and correcting the phrase described below. 

 on page 10, 7: “previous research” would be probably to In the previous research.

Author Response

Reviewer #2

The present study has demonstrated the protective effect of Kombucha tea (KT) against the both of lipid accumulation and lipotoxicity, in vivo and vitro.  Although It has been already shown the protective effect of KT against non-alcoholic fatty liver disease (NAFLD), in there, KT was shown to redress the dysbiosis induced by the methionine/choline-deficient diets.  This study has exhibited KT can intervene the hedgehog involving in non-alcoholic steatohepatitis (NASH). Why doesn't this author make the description about the hedgehog in Abstract and Discussion. The description in there will give new and distinctive impact for this manuscript. 

Although we did not explain Hh-related data in abstract, we described Hh signaling and the Hh-related results from line 5 to line 21 in page 12. In the revised manuscript, we briefly described Hh signaling in abstract because of word limitation (200), as you requested; “With reduction of hedgehog signaling, inflammation and fibrosis also declined in the M+K group”.

Therefore, this paper would be suitable for publication as an original article after making addition of the above matters and correcting the phrase described below.

on page 10, 7: “previous research” would be probably to In the previous research.

Thank you for your helpful comments. We corrected them.

In the manuscript, we clearly wrote that “The present study reveals the effect of KT in an experimental NASH model. In the previous research, ~~~~”. However, it seems that some words are missing in the process of changing to a form requested by IJMS journal. Hence, after changed the form of the revised manuscript to IJMS form, we carefully checked it to avoid repeating the same mistake.

Round 2

Reviewer 1 Report

Comments

Authors have revised their paper according to the suggestions of the reviewer. Figures were improved. The English has become much better but there are still minor changes to be done by the editorial team. The paper now could be accepted for publication.